# Physical function, daily activities, and spinal pain in the elderly: A cross-sectional study

Paulina Kowalewska[1]*, Małgorzata Wójcik[2], Aleksandra Banaszak[1], Kacper Bernatowicz[1,3], Mateusz Motyl[1], Patrycja Wołodźko[1], Matylda Sauermann[1], Maciej Wołczek[1], Eryk Pietruszak[1], Bartosz Aniśko[2]

1 Physio Research Review Student Research Association, Poznań University of Physical Education, Faculty of Sport Sciences, Gorzów Wielkopolski, Poland, 2 Department of Physiotherapy, Poznań University of Physical Education, Faculty of Sport Sciences, Gorzów Wielkopolski, Poland, 3 Conocimiento Student Research Association, Poznań University of Physical Education, Faculty of Sport Sciences, Gorzów Wielkopolski, Poland

* pkowalewska175@gmail.com

## Abstract

Population aging is associated with progressive functional decline and a rising prevalence of chronic spinal pain, representing a major challenge for public health and geriatric care.

### Objective

To evaluate the relationships between spinal pain, functional mobility, balance, and physical activity level in older adults.

### Methods

A total of 28 community-dwelling seniors (25 women, 3 men; mean age 70.8±5.1 years) participated in the study. Functional mobility was assessed using the Timed Up and Go (TUG) test and reaction time. Postural balance was measured using a force platform. Disability was evaluated with NDI and ODI, and physical activity with IPAQ.

### Results

Significant correlations were found between age and TUG (r=0.61, p<0.001), and between Neck Disability Index (NDI) and postural sway area (r=0.45, p=0.016). No significant relationship was observed between pain and physical activity levels.

### Conclusion

Spinal pain and age are significantly associated with functional decline and impaired balance in older adults. These findings suggest that assessing cervical disability and postural stability should be integral to the geriatric evaluation, as they may identify individuals at higher risk of mobility limitations.

**Data availability statement:** All files are available in the public repository at Repository link: https://doi.org/10.18150/Z6ELNF (DOI: 10.18150/Z6ELNF).

**Funding:** The author(s) received no specific funding for this work.

## 1. Introduction

Life expectancy in Poland has been gradually increasing, accompanied by an extension of the healthy life span [1]. Over the past seven decades, the average life expectancy of the global population has risen by 26 years [2]. As a result, contemporary medicine focuses not only on prolonging life but also on improving its quality, particularly by supporting both physical and mental well-being, and by promoting the maintenance of good health for as long as possible [3]. Older adults are at greater risk of developing chronic diseases. Among individuals aged 60, 73% suffer from at least one chronic condition, and in those over 70 years of age this percentage increases to as much as 84% [4]. The decline in physical capacity among older adults particularly affects the musculoskeletal system and motor function, leading to reduced independence in daily life as well as the onset of pain. Some of the major age-related musculoskeletal problems include loss of muscle strength, function, and mass (sarcopenia), decreased bone mass (osteoporosis), and degenerative joint diseases (e.g., osteoarthritis), which, if left untreated, can deprive older adults of their independence [5–7]. These conditions contribute to an increased risk of falls and fractures, making seniors far more vulnerable to mobility limitations and disability [8,9]. A meta-analysis conducted in 2022 reported that 26.5% of older adults worldwide had experienced at least one fall [10]. According to the World Health Organization, falls are the second leading cause of death from unintentional injuries worldwide, with approximately 37.3 million falls each year severe enough to require medical attention [3]. Physical fitness can be achieved through regular exercise as well as through spontaneous physical activity. It provides resilience against potentially harmful behavioral and metabolic consequences of stressful events and helps prevent many chronic diseases by exerting both physical and psychological benefits on the body [11,12]. Physical inactivity increases the risk of developing major non-communicable diseases, such as cardiovascular diseases, type 2 diabetes, obesity, cancer, and musculoskeletal disorders. Furthermore, it contributes to higher mortality rates among patients affected by these conditions [13]. Maintaining physical fitness in older age is particularly important for preserving independence for as long as possible. Physical activity promotes healthy aging and protects against the decline of physical and cognitive functions [14]. A 6-year longitudinal study on aging demonstrated that older adults maintaining moderate physical activity had a lower risk of all-cause mortality compared to their low-activity counterparts. Furthermore, individuals who initially exhibited high levels of activity that significantly decreased over time faced a mortality risk comparable to that of the low-activity group [15]. However, despite these benefits, pain remains highly prevalent in older adults and often represents a major barrier to maintaining regular physical activity. Pain is defined as an unpleasant, subjective sensory and emotional experience [16]. It is a complex phenomenon that arises from tissue-damaging stimuli and may also be caused by neurological injury, such as stroke, peripheral neuropathy, spinal radiculopathy, or cranial neuralgias. Pain is modulated by higher centers of the nervous system [17]. It is typically associated with injury or a pathophysiological process that evokes this unpleasant experience [18]. In older adults, pain most commonly takes a chronic form [19]. Seniors experience both acute and chronic pain conditions, including neuropathic pain, nonspecific

joint pain, headache, as well as pain in the neck, back, hand, hip, knee, and foot [20–33]. The most frequently reported site of chronic pain is the back (approximately 5–45%), followed by the neck and hip (about 20%), and then the knee (around 18%). However, in terms of overall prevalence, the highest proportions are found for chronic neuropathic pain (10–52%) and chronic nonspecific joint pain (around 40%) [34]. Neck pain ranks as the fourth leading cause of disability, whereas low back pain constitutes the primary cause of productivity loss worldwide [35]. An analysis of prevalence trends between 1990 and 2021 revealed a slight decrease in low back pain, contrasting with an upward trend observed for neck pain. Furthermore, both conditions were more prevalent in women than in men [36]. In 2020, neck pain affected 203 million individuals globally, while low back pain impacted as many as 619 million people [37,38]. With regard to sex differences, chronic spinal pain is statistically more common in women, with its peak prevalence observed in middle age [36]. In Poland, according to Ministry of Health data from 2014, the annual prevalence of chronic cervical spine pain was 21% among women and 13% among men. For lower back pain, higher rates were also noted in women (28.4%) compared to men (21.2%) [39]. Spinal pain is multifactorial, which makes its elimination particularly challenging. Ergonomic factors include physical activity, force exertion, repetitive movements, and improper posture. Individual factors comprise age, genetic predisposition, body mass index, and history of musculoskeletal pain. Behavioral determinants include smoking and physical activity level, while psychosocial factors encompass job satisfaction, anxiety, stress, and depression [40].

In the context of chronic spinal pain and multimorbidity in older adults, aging is characterized by multifactorial, functional, and morphological changes that ultimately lead to a decline in physical fitness [41]. Low levels of physical fitness may increase the risk of deterioration in both functional and cognitive health [42,43], the development of chronic diseases [44,45], sarcopenia [46], and premature mortality [47]. Older adults typically present with reduced aerobic endurance and impaired balance—two key components of physical fitness—which contribute to a lower quality of life [48]. To counteract the negative consequences of aging, strategies have been developed that emphasize increasing physical activity, which in turn improves physical fitness. A study by Bamini Gopinath demonstrated a positive association with the concept of multi-dimensional successful aging during a ten-year follow-up [49]. Against this background, the aim of this study was to evaluate the functioning and physical fitness of older adults experiencing spinal pain, a condition that often limits daily activities and increases dependence on caregivers [50]. Chronic spinal pain may also contribute to depression and social isolation, which further reduce quality of life [51]. In addition, it can impair balance and proprioception, thereby elevating the risk of falls that frequently lead to serious injuries in this population [52]. Considering these challenges, systematic research is essential for developing effective physiotherapy programs. Given the global trend of population aging, the burden of spinal pain is expected to increase [53]. Therefore, it is crucial not only to investigate effective treatment methods but also to identify strategies that can improve the overall quality of life in older adults.

In this context, we formulated the following hypotheses. We hypothesized that higher levels of spinal pain (particularly cervical) would be significantly correlated with decreased functional mobility, impaired postural balance, and lower physical activity levels in older adults. We further expected that greater physical activity levels (measured in MET units) would correlate with better outcomes in physical fitness tests and with lower spinal pain intensity. In contrast, longer sitting time was anticipated to be linked to greater pain severity and poorer functional performance. We also hypothesized that higher hip muscle strength in flexion and extension would be associated with longer step length and better Timed Up and Go test results. Finally, we expected that advancing age would negatively influence physical fitness outcomes, step length, and reaction time. Despite the high prevalence of spinal pain in the elderly, there is a paucity of studies directly analyzing how specific pain locations (cervical vs. lumbar) correlate with objective biomechanical parameters of balance and functional mobility in this population. Most existing research relies on subjective questionnaires alone, leaving the sensorimotor mechanisms insufficiently explored.

## 2. Materials and methods

This study was designed as a cross-sectional observational study and followed the STROBE guidelines for reporting observational studies. A total of 28 individuals participated in the study (25 women and 3 men; mean age 70.8±5.10

years). Participants were recruited from local senior clubs in Gorzów Wielkopolski via printed flyers and oral announcements. The recruitment process lasted from July to September 2024. Volunteers who expressed interest were screened for eligibility based on the inclusion and exclusion criteria. Prior to enrollment, all participants provided written informed consent for participation, the processing of personal data, and the anonymous publication of results.

The study was conducted in accordance with the principles of the Declaration of Helsinki. All participants were informed about the purpose and procedures of the study and provided written informed consent to participate. The study protocol was approved by the Bioethics Committee at the Poznan University of Medical Sciences (decision no. [440/24]).

The study was conducted in a human movement laboratory. Participants attended the laboratory between 8:00 and 11:00 a.m. In the first stage, they were asked to complete questionnaires assessing physical fitness, physical activity levels, and pain-related complaints. Subsequently, each participant underwent a series of physical fitness tests.

The inclusion criteria were: (1) age 60 years or older; and (2) good general health status allowing for safe ambulation on a treadmill and performance of functional tests.

The exclusion criteria were: (1) musculoskeletal dysfunctions preventing independent walking; (2) history of lower limb fractures or surgery within the last 6 months; (3) presence of lower limb prostheses; (4) severe balance disorders or neurological conditions impairing independent mobility and reaction to stimuli; and (5) neurodegenerative diseases significantly affecting cognitive functions (preventing understanding of instructions).

## 2.1. Questionnaires

Only standardized questionnaires widely used in scientific research were applied in this study. The Barthel Index is a standardized tool for assessing functional independence, commonly used in both clinical research and medical practice to determine the degree of self-sufficiency in performing basic activities of daily living. The scale evaluates patient independence across ten categories related to self-care and mobility, including feeding, bathing, personal hygiene, dressing, bladder and bowel control, transfers, walking, and stair climbing. Each activity is assigned a specific score depending on the patient's level of independence, with a maximum of 100 points indicating full functional independence, and a score of 0 indicating total dependence. Interpretation of the results allows classification into three groups: completely dependent (0–20 points), moderately dependent (21–60 points), and functionally independent (≥61 points) [54].

The Neck Disability Index (NDI) is one of the most widely used tools for assessing disability resulting from neck pain. It is a standardized questionnaire designed to evaluate the impact of pain on patients' daily functioning. The instrument consists of 10 items addressing key aspects of life, including pain intensity, self-care (e.g., dressing, hygiene), lifting objects, reading, headaches, concentration, work activities, driving, sleeping, and recreational activities. Each item is scored on a 0–5 scale, where 0 indicates no difficulty and 5 reflects complete limitation of the given activity. The total score ranges from 0 to 50 points and is converted into a percentage disability index, which determines the level of impairment associated with neck pain. Higher scores correspond to greater functional limitations. The NDI is widely used in both clinical practice and research to monitor treatment progress and to evaluate the effectiveness of various therapeutic interventions in patients with cervical spine dysfunction [55–59].

The Activities of Daily Living (ADL) questionnaire assesses a patient's ability to perform fundamental tasks essential for independent functioning. It consists of six categories: personal hygiene (bathing), dressing, feeding, bladder and bowel control, transfers, and toileting. In contrast, the Instrumental Activities of Daily Living (IADL) scale evaluates more complex life skills that require cognitive and organizational abilities. It covers eight main areas: using the telephone, managing finances, shopping, meal preparation, housekeeping, medication management, transportation, and performing domestic chores. The assessment of ADL and IADL is an important component of the diagnostic process and therapeutic planning, enabling the monitoring of patient progress and the adjustment of treatment strategies [60].

The International Physical Activity Questionnaire (IPAQ) is one of the most commonly used tools for assessing physical activity in populations worldwide. It is available in two versions: the short form and the long form. The long version

(IPAQ-L) allows for a more detailed evaluation of various aspects of physical activity by taking into account different contexts in which it is performed. The questionnaire consists of 27 items divided into five main domains of physical activity carried out during the past 7 days: work-related activity (e.g., walking, sedentary work, physically demanding tasks), walking and cycling for transportation, household and gardening activities (e.g., cleaning, gardening), leisure-time, sports, and exercise activities (e.g., strength training, running, cycling, swimming), and time spent sitting (e.g., at work, at home, during study, or leisure). The results are expressed in Metabolic Equivalent of Task (MET-min/week), calculated based on the time and intensity of physical activity. Three levels of physical activity are distinguished: low (<600 MET-min/week) – little or no physical activity; moderate (600–3000 MET-min/week) – activity meeting health recommendations; and high (>3000 MET-min/week) – vigorous physical activity. The IPAQ-L is widely used in epidemiological studies, lifestyle analyses, public health assessments of physical activity, and in monitoring changes in population movement behaviors [61–63].

The Oswestry Disability Index (ODI) is one of the most commonly used tools for assessing disability related to lower back pain. It consists of 10 items addressing pain intensity and its impact on daily activities such as walking, sitting, self-care, sleeping, and social activities. Each item is scored on a 0–5 scale, and the total score is converted into a percentage scale of disability (0–100%). Interpretation includes five levels: minimal disability (0–20%), moderate disability (21–40%), severe disability (41–60%), crippling disability (61–80%), and complete disability (81–100%). The ODI is widely applied in diagnostics, rehabilitation, and clinical research, allowing for the evaluation of treatment progress and the effectiveness of therapeutic interventions in patients with chronic back pain [64–66].

## 2.2 Assessment of physical fitness and body composition

After completing the questionnaires, participants proceeded to the next stage of the study. The first test was a body composition analysis, performed using a professional bioelectrical impedance analyzer (Tanita-980 M, Tanita Corporation, Tokyo, Japan). The device is medically certified, complies with NAWI and CLASS III standards for medical weighing instruments, and holds the CE0122 certification confirming compliance with the Medical Device Directive 93/42/EEC. Prior to the measurement, participants were asked to remove all metal objects and to take off their shoes and socks to ensure the accuracy of results. They then stood on the analyzer in designated positions while holding the hand electrodes. During the 30-second assessment, participants were required to remain completely still to guarantee precise readings. The analyzed parameters included total body mass, muscle mass, fat mass, and bone mass, allowing for a comprehensive evaluation of body composition [67].

Grip strength in the left and right upper limb was assessed using an electronic hand dynamometer (Saehan, Belgium). Participants stood upright with their arms relaxed alongside the body and elbows fully extended. On the examiner's signal, they were instructed to maximally squeeze the dynamometer with one hand while maintaining the prescribed body position. The task was to generate the maximum possible grip force. Results were expressed in kilograms (kg), providing an objective measure of hand and forearm muscle strength [68].

Hip flexion and extension strength was assessed using a multifunctional dynamometer (Meloq Devices, Sweden). The device was attached to a chain fixed to a wall-mounted handle to ensure measurement stability. The other end of the attachment system was fastened to the participant's thigh, precisely 10 cm above the knee joint line, allowing accurate transmission of the force generated by the hip muscles. During the measurement, participants stood at an optimal distance that enabled a full range of motion in both flexion and extension, while exerting maximal possible effort. Given the age of the participants, a support device was provided so they could hold on to it for increased stability and safety during the test. Muscle strength results were expressed in kilograms (kg), allowing for an objective analysis of participants' functional capacity [69,70].

Static balance was assessed using a dual-plate posturographic platform (Koordynacja, Poland), which enables precise measurement of postural stability. During the test, participants stood symmetrically on the measurement surface while barefoot to eliminate the potential influence of footwear on stability. The assessment lasted 30 seconds, during which participants

were instructed to maintain an immobile body position and minimize unnecessary movements. They were also asked to focus their gaze on a designated point at eye level to limit the influence of visual control on balance. The primary parameter analyzed was the sway area (expressed in mm²), reflecting overall postural stability. Higher values indicated greater center of pressure (COP) displacement, which is interpreted as reduced ability to maintain static balance [71–73].

Step length was measured using an advanced gait analysis treadmill (Zebris FDM-THQ, Zebris Medical GmbH, Germany), which allows for precise assessment of gait parameters. Before the test, participants received detailed instructions on how to walk on the treadmill and were informed about safety procedures. They then stepped onto the treadmill barefoot and initially held the safety handrail. The treadmill speed was gradually increased to match the participant's natural walking pace, typically within the range of 2–3 km/h. After adapting to the treadmill motion, participants were instructed to release the handrail and walk freely. The actual measurement, performed once a steady walking rhythm was achieved, lasted 30 seconds. The final results included the step length of both the right and left leg, expressed in centimeters (cm) [74,75].

Functional mobility was assessed using the Timed Up and Go (TUG) test. Participants were instructed to stand up from a standard chair, walk 3 meters as quickly but safely as possible, turn around, walk back, and sit down. The time required to complete the task was recorded in seconds. This test evaluates dynamic balance and functional mobility, rather than isolated gait speed. The test involved standing up from a seated position, walking a distance of 3 meters, turning around a designated obstacle, returning along the same path, and sitting back down on the chair. Participants were instructed to complete the task as quickly as possible without transitioning to a run. The time was measured from the examiner's "start" command until the participant had fully resumed the seated position at the end of the trial. The results were expressed in seconds (s), with shorter times indicating better overall functional mobility and motor agility. [76–78].

Reaction time and visuomotor coordination were measured using the WittySem semaphore system. The semaphores were mounted on a special stand at a fixed height, regardless of the participant's stature. Each test trial began with a 3-second countdown. The participant's task was to identify the small green letter "a" displayed among four semaphores showing various digits and letters in different colors, and to move their hand toward it as quickly as possible. The task was repeated 20 times. The entire test was performed three times, and the best result was recorded. Outcomes were expressed in seconds (s) [79].

All instruments used in this study demonstrate well-established psychometric properties in older adult populations. The Neck Disability Index (NDI) shows strong construct validity and excellent internal consistency (Cronbach's α = 0.87–0.92), with high test–retest reliability (ICC = 0.89–0.98) and documented responsiveness in patients with mechanical neck disorders [57–59]. The Oswestry Disability Index (ODI) also demonstrates good internal consistency (α = 0.81–0.86) and test–retest reliability (ICC = 0.84–0.94) in older adults with low back pain, with strong cross-cultural stability [64,66]. The International Physical Activity Questionnaire (IPAQ) shows acceptable reliability in older adults (ICC ≈ 0.74–0.75), although its tendency to overestimate activity levels is well documented [61]. The Timed Up and Go test (TUG) exhibits excellent reliability in community-dwelling older adults (ICC = 0.96–0.99) and strong predictive validity for fall risk and functional mobility [77,78]. Postural sway measures derived from stabilometric platforms demonstrate good test–retest reliability (ICC = 0.70–0.89) and strong validity for assessing balance deficits in older adults [72,73]. Hip extensor strength assessed using fixed and hand-held dynamometry has shown high reliability (ICC = 0.93–0.99 and ICC > 0.80, respectively) [70]. Reaction-time assessment using wireless LED-based systems is validated in older adults and demonstrates high test–retest reliability (ICC > 0.80), supporting its suitability for neuromotor performance testing in geriatric populations [79]. Normative step-length characteristics and their measurement reliability in older adults are well established in the literature [75].

## 2.3. Statistical analysis

All statistical analyses, calculations, and visualizations were performed using Jamovi and RStudio software. A post-hoc power analysis indicated that a sample size of 28 provided adequate power (>80%) to detect strong correlations (Spearman's rho ≥ 0.52) at a two-tailed significance level of 0.05. The Shapiro–Wilk test was applied to assess the normality of

data distribution, while Pearson's and Spearman's tests were used to evaluate correlations between variables, as appropriate. The level of statistical significance was set at $p < 0.05$ [80,81]. We acknowledge the importance of adjusting for covariates such as age and muscle mass. However, given the relatively small sample size (N = 28), performing multivariate regression analyses or ANCOVA would carry a high risk of model overfitting and insufficient statistical power. Adhering to the rule of thumb of 10–15 subjects per predictor variable, our sample size only allows for robust bivariate analyses. Therefore, we restricted our statistical approach to Spearman's correlations to ensure the validity of the results, while acknowledging this limitation in the Discussion section.

## 3. Results

In total, 28 community-dwelling older adults (25 women, 3 men) participated in the study, with a mean age of 70.8 ± 5.1 years. On average, participants were classified as obese (BMI 31 kg/m²) and reported low-to-moderate levels of cervical and lumbar disability (mean NDI 7.14, ODI 11.75). Functional mobility and balance parameters (TUG, step length, postural sway) as well as physical activity levels (IPAQ) showed substantial inter-individual variability (Table 1).

**Table 1. Demographic characteristics and clinical outcomes of the study participants (N = 28).**

| Variable | Mean ± SD | Range (Min-Max) |
|---|---|---|
| **Demographics** | | |
| Age (years) | 70.79 ± 5.10 | 61.0 - 80.0 |
| Height (cm) | 156.82 ± 6.64 | 148.0 - 178.0 |
| Body Mass (kg) | 76.54 ± 12.88 | 55.7 - 112.8 |
| BMI (kg/m²) | 31.00 ± 3.73 | 24.1 - 38.2 |
| Muscle Mass (kg) | 45.03 ± 8.20 | 36.1 - 72.2 |
| Fat Mass (kg) | 29.22 ± 5.95 | 19.1 - 43.0 |
| **Pain & Disability** | | |
| NDI (Neck Disability Index) | 7.14 ± 5.35 | 0.0 - 23.0 |
| ODI (Oswestry Disability Index) | 11.75 ± 9.84 | 0.0 - 33.0 |
| **Functional Tests** | | |
| TUG Test (s) | 8.46 ± 2.63 | 5.8 - 17.0 |
| Reaction Time (s) | 31.90 ± 8.55 | 23.3 - 61.0 |
| Step Length Left (cm) | 33.40 ± 10.15 | 16.0 - 52.0 |
| Step Length Right (cm) | 35.63 ± 9.98 | 14.0 - 53.0 |
| Postural Sway Area (mm²) | 529.61 ± 352.83 | 112.0 - 1260.0 |
| Hip Ext. Strength L (kg) | 20.19 ± 8.74 | 4.4 - 44.0 |
| Hip Ext. Strength R (kg) | 21.35 ± 8.43 | 4.4 - 40.5 |
| **Physical Activity** | | |
| IPAQ (MET-min/week) | 9099.71 ± 8418.92 | 750.0 - 31422.0 |
| Sitting Time (min/week) | 1395.00 ± 527.60 | 420.0 - 2520.0 |

Note: BMI = Body Mass Index; NDI = Neck Disability Index; ODI = Oswestry Disability Index; TUG = Timed Up and Go test; IPAQ = International Physical Activity Questionnaire.

When analyzing the impact of cervical (NDI) and lumbar (ODI) spine pain on the physical performance of participants, Spearman's nonparametric test was applied, as the data did not meet the assumptions of normal distribution. Spearman's correlation analysis revealed a moderate positive correlation between NDI and BALANCE (r = 0.45, p = 0.016).

In contrast, ODI scores did not show a significant correlation with balance (r = 0.22, p = 0.267). Lumbar spine pain did not correlate with most physical fitness test outcomes. The only exception was hip extension strength in the right leg, where higher ODI scores were associated with lower muscle strength in hip extension (r = −0.437, p = 0.020).

The analysis of the relationship between age and step length in both the left and right leg revealed significant negative correlations (left leg: r = −0.50, p = 0.007; right leg: r = −0.55, p = 0.003; Table 2).Spearman's correlation analysis showed a significant positive association between age and the time required to complete the Timed Up and Go test (r = 0.61, p < 0.001; Table 2), with longer TUG times observed in older participants.Spearman's correlation analysis revealed a significant positive relationship between age and reaction time (r = 0.41, p = 0.030), with longer reaction times observed at higher ages.Spearman's correlation analysis demonstrated a significant positive association between age and time spent sitting (r = 0.47, p = 0.012), with higher age associated with greater weekly sitting time.Spearman's correlation analysis revealed a significant negative relationship between MET and reaction time (r = −0.57, p = 0.0017), with higher MET values associated with shorter reaction times.Physical activity level did not significantly affect other variables, including cervical and lumbar spine pain (p > 0.05). Hip flexion and extension strength showed no significant relationship with participants' step length. However, hip extension strength in both the right and left leg was significantly correlated with Timed Up and Go performance (r = −0.43, p = 0.023; r = −0.47, p = 0.012).Time spent sitting showed one additional significant association that was not observed in other physical fitness tests. Spearman's correlation analysis revealed that longer sitting time was associated with greater postural sway (r = 0.41, p = 0.022). Apart from this relationship and the reduction in right hip extension strength (r = −0.42, p = 0.027), sitting time did not significantly affect the remaining physical fitness outcomes.Overall, the correlation pattern presented in Table 2 indicates that cervical disability and age are the variables most consistently associated with poorer balance, slower mobility, shorter step length and longer reaction times, whereas physical activity levels showed a more selective relationship, mainly with neuromotor performance (reaction time).

## 4. Discussion

The present study evaluated the correlations between physical fitness indicators in older adults and their levels of physical activity, advancing age, and cervical and lumbar spine pain. Higher levels of pain in these spinal regions were negatively associated with components of physical fitness essential for daily functioning, such as postural control and lower limb strength. This pattern suggests that older adults reporting greater cervical and lumbar spinal pain tend to present with reduced functional capacity, which may limit their independence in everyday activities.

**Table 2. Spearman's rank correlation coefficients (r) between age, pain, and functional parameters.**

|  | Age | NDI | ODI | TUG (s) | RT (s) | Sway (mm²) | S-L L (cm) | S-L R (cm) | Hip Str. L (kg) | Hip Str. R (kg) | IPAQ (MET) | ST (min/wk) |
|---|---|---|---|---|---|---|---|---|---|---|---|---|
| Age | – | −0.05 | −0.04 | **0.61*** | **0.41*** | 0.16 | **−0.50*** | **−0.55*** | −0.18 | −0.35 | −0.04 | **0.47*** |
| NDI | −0.05 | – | **0.40*** | 0.10 | −0.10 | **0.45*** | −0.09 | 0.04 | −0.03 | −0.25 | 0.08 | 0.11 |
| ODI | −0.04 | **0.40*** | – | −0.07 | −0.10 | 0.22 | −0.08 | 0.14 | −0.21 | **−0.44*** | 0.04 | 0.26 |
| TUG (s) | **0.61*** | 0.10 | −0.07 | – | **0.60*** | 0.12 | −0.17 | −0.30 | **−0.47*** | **−0.43*** | −0.13 | 0.14 |
| RT (s) | **0.41*** | −0.10 | −0.10 | **0.60*** | – | 0.05 | **−0.41*** | **−0.47*** | −0.30 | −0.35 | **−0.57*** | 0.25 |
| Sway (mm²) | 0.16 | **0.45*** | 0.26 | 0.12 | 0.05 | – | −0.02 | 0.09 | −0.03 | −0.36 | −0.03 | **0.41*** |
| S-L L (cm) | **−0.44*** | −0.09 | −0.08 | −0.17 | **−0.41*** | −0.02 | – | **0.81*** | 0.28 | 0.36 | −0.01 | −0.27 |
| S-L R (cm) | **−0.56*** | 0.04 | 0.14 | −0.30 | **−0.47*** | 0.09 | **0.81*** | – | 0.31 | 0.20 | 0.01 | −0.21 |
| Hip Str. L (kg) | −0.18 | −0.03 | −0.21 | **−0.47*** | −0.30 | −0.03 | 0.28 | 0.31 | – | **0.76*** | −0.13 | −0.12 |
| Hip Str. R (kg) | −0.35 | −0.25 | **−0.44*** | **−0.43*** | −0.35 | −0.36 | 0.36 | 0.20 | **0.76*** | – | 0.06 | **−0.42*** |
| IPAQ (MET) | −0.04 | 0.08 | 0.04 | −0.13 | **−0.57*** | −0.03 | −0.01 | 0.01 | −0.13 | 0.06 | – | −0.29 |
| ST (min/wk) | **0.47*** | 0.11 | 0.26 | 0.14 | 0.25 | **0.41*** | −0.27 | −0.21 | −0.12 | **−0.42*** | −0.29 | – |

Values in bold with an asterisk (*) indicate statistically significant correlations (p < 0.05).

NDI = Neck Disability Index; ODI = Oswestry Disability Index; TUG = Timed Up and Go test; RT = Reaction Time; Sway = Postural Sway Area; S-L L/R = Step Length (Left/Right); Hip Str. L/R = Hip Extension Strength (Left/Right); IPAQ = International Physical Activity Questionnaire; ST = Sitting Time.

Advancing age was associated with less favourable gait parameters, with step length becoming shorter, mobility test results poorer, and reaction times longer. Age-related decline in basic locomotor abilities not only increases reliance on others for daily tasks but also diminishes the sense of independence and autonomy among older adults. In contrast, seniors with higher levels of physical activity demonstrated faster reaction times, which in the context of daily life may play a protective role against falls, a common and serious problem in this age group. Conversely, participants with more sedentary lifestyles exhibited reduced strength in the hip extensor muscles, which may compromise stability and dynamic movement.

Many authors have also confirmed the association between lumbar spine pain and reduced physical fitness in older adults. Disability resulting from pain is often dependent on its intensity, while higher levels of physical activity have been shown to reduce pain in these regions [82–84]. A cross-sectional study conducted in Singapore further corroborates the significant impact of chronic low back pain on both physical and mental health. Chronic low back pain was associated with reduced physical function, greater functional limitations, depressive symptoms, and generally lower health-related quality of life [85].

Research on the correlation between cervical spine pain and impaired balance in older adults has also demonstrated significant associations.

The significant positive correlation observed in our study between neck disability (NDI) and postural sway area (r = 0.45) aligns with the sensorimotor deficit model proposed in recent literature. Reddy et al. (2023) demonstrated that age-related decline in cervical proprioception is strongly associated with impaired functional mobility and limits of stability [86]. Pain acts as a compounding factor in this process; Uthaikhup et al. (2012) found that older adults with neck pain exhibit significantly greater postural instability compared to pain-free controls due to altered afferent input from cervical mechanoreceptors [87]. Our findings are consistent with this proposed mechanism and suggest that even moderate cervical disability may be accompanied by alterations in the integration of vestibular and visual signals required for maintaining balance, which may contribute to an increased risk of falls as highlighted by Kendall et al. (2018) [88].

Pain in the cervical region has been reported to be associated with reduced stability and balance, and may also be associated with decreased strength of the back extensors [89,90]. Although most suggest that chronic spinal pain is associated with poorer fitness, mobility, and balance in seniors, it is important to consider other contributing factors such as physiological aging, declining muscle strength, and comorbidities. Nonetheless, both the high prevalence of spinal pain in the oldest age groups and the growing body of research underscore the importance of identifying strategies to reduce chronic spinal pain.

In addition to the relationships between cervical disability and postural sway, several other correlational patterns observed in our study are consistent with the broader geriatric literature. Higher age was associated with shorter step length, slower performance in the Timed Up and Go test and longer reaction times, reflecting the well-documented age-related decline in locomotor efficiency and neuromotor processing in older adults. At the same time, higher levels of self-reported physical activity were related to shorter reaction times, whereas prolonged sitting time was associated with greater postural sway and reduced hip extensor strength. Taken together, these findings suggest that, in community-dwelling older adults, spinal pain, aging and sedentary behaviour tend to co-occur with a less favourable profile of physical fitness, while higher levels of physical activity are linked primarily with better neuromotor performance.

Similarly, the negative impact of aging on gait parameters was demonstrated in a 2016 study, where older adults exhibited shorter, wider, and slower steps compared to younger individuals, which may represent a compensatory strategy to prevent falls [91]. Gamwell (2022) also showed that both aging and physical activity level affect gait quality. Higher levels of physical activity were correlated with greater strength of the ankle joint muscles, which in turn contributed to improved gait stability [92]. The tendency for step length to shorten and gait parameters to deteriorate with advancing age has been consistently reported by numerous authors. The primary mechanisms are thought to involve declines in muscle strength, particularly in the ankle and hip joints. Older adults who engage in higher levels of physical activity achieve better

outcomes in terms of muscle strength, mobility, and gait stability. Therefore, greater attention should be paid to promoting daily physical activity in seniors, as it can enhance their safety by reducing the risk of falls and simultaneously support greater independence [93,94].

Many authors have also reported that older adults demonstrate longer reaction times compared to younger individuals [95–97]. Our findings are consistent with this relationship and further suggest that more physically active seniors exhibit shorter reaction times. Similar results were obtained by Liu, who demonstrated that older adults with higher levels of physical activity during leisure time and household tasks responded more quickly to stimuli. However, no significant differences were observed between seniors with "moderate" and "high" activity levels, indicating that even moderate physical activity is sufficient to improve functional performance and quality of life in older adults [98].

The obtained results were generally consistent with our initial hypotheses. Higher scores in the NDI and ODI questionnaires were associated with poorer balance and reduced muscle strength, supporting the notion that greater spinal disability tends to co-occur with functional limitations. Moreover, a higher level of physical activity was related to better performance in fitness tests, particularly shorter reaction times, while longer sitting time correlated with decreased hip extensor strength, and its association with disability measures was weaker and did not reach statistical significance. Significant associations were also observed between muscle strength, step length, and mobility as assessed by the TUG test. As expected, higher age was associated with less favourable values of all key physical fitness parameters, including reaction time, step length, and mobility. The only weaker associations were those between physical activity and pain intensity, which may be attributed to the relatively small sample size. These patterns should be interpreted as associations rather than causal effects, given the cross-sectional design and modest sample size of the study. Furthermore, the negative correlations between hip extensor strength and TUG time, as well as the associations of longer sitting time with greater postural sway and weaker hip muscles, underline the central role of lower limb strength and sedentary behaviour in shaping functional mobility in this age group. The findings of this study have important practical implications. First, the observed associations between spinal pain, physical activity levels, and functional performance suggest that comprehensive assessment of older adults should include both disability questionnaires (NDI, ODI) and performance-based tests (e.g., TUG, step length, reaction time). These complementary measures can help identify individuals at higher risk of functional decline. Second, the demonstrated relationships between low physical activity, prolonged reaction time, and reduced mobility underscore the need to implement targeted physiotherapy and exercise programs. Even regular, moderate-intensity activity may contribute to improved neuromuscular function, reduction of spinal pain, and enhanced postural stability, ultimately supporting independence in daily living. Third, the results highlight the importance of reducing sedentary behavior among older adults. Interventions aimed at decreasing sitting time and introducing regular movement breaks may help maintain muscle strength and functional abilities. Finally, individualized exercise programs focusing on strengthening the hip and trunk muscles, improving balance, and increasing overall physical activity may play a particularly beneficial role in reducing disability and fall risk in seniors with spinal pain.

## Conclusions

Spinal pain, particularly in the cervical region, and advancing age are significantly associated with functional decline and impaired postural balance in older adults. Our findings highlight that even moderate neck disability (NDI) correlates with increased postural sway, suggesting a disruption in sensorimotor control. Consequently, routine geriatric assessment should include evaluation of cervical function and balance to identify individuals at higher risk of mobility limitations and falls. Future longitudinal studies are needed to determine causal relationships and test targeted interventions.

## Limitations

This study has several limitations that should be considered when interpreting the results. First, the sample size was relatively small, and the study group was predominantly female, which limits the generalizability of the findings to the broader

population of older adults. Another limitation was the cross-sectional design, which allowed for the analysis of associations but did not permit conclusions regarding causal mechanisms underlying the observed relationships. In addition, physical activity was assessed using a self-reported questionnaire (IPAQ), which may be subject to bias arising from participants' subjective evaluation. It should also be noted that the study did not account for the influence of comorbid chronic conditions or ongoing pharmacotherapy, both of which may have affected functional performance and pain perception. Future research should therefore involve larger and more diverse samples, include objective measures of physical activity, and consider clinical as well as sociodemographic factors. The small sample size also precluded multivariate adjustments for potential confounders such as age or muscle mass.

## Author contributions

**Conceptualization:** Paulina Kowalewska, Małgorzata Wójcik, Eryk Pietruszak, Bartosz Aniśko.

**Data curation:** Paulina Kowalewska, Małgorzata Wójcik, Aleksandra Banaszak, Kacper Bernatowicz, Mateusz Motyl, Patrycja Wołodźko, Eryk Pietruszak, Matylda Sauermann, Maciej Wołczek, Bartosz Aniśko.

**Formal analysis:** Małgorzata Wójcik, Bartosz Aniśko.

**Investigation:** Paulina Kowalewska, Małgorzata Wójcik, Bartosz Aniśko.

**Methodology:** Paulina Kowalewska, Małgorzata Wójcik, Eryk Pietruszak, Bartosz Aniśko.

**Project administration:** Paulina Kowalewska, Małgorzata Wójcik, Bartosz Aniśko.

**Resources:** Paulina Kowalewska, Małgorzata Wójcik, Bartosz Aniśko.

**Software:** Paulina Kowalewska, Małgorzata Wójcik, Bartosz Aniśko.

**Supervision:** Małgorzata Wójcik, Bartosz Aniśko.

**Validation:** Małgorzata Wójcik, Bartosz Aniśko.

**Visualization:** Paulina Kowalewska, Małgorzata Wójcik, Bartosz Aniśko.

**Writing – original draft:** Paulina Kowalewska, Małgorzata Wójcik, Aleksandra Banaszak, Kacper Bernatowicz, Mateusz Motyl, Patrycja Wołodźko, Eryk Pietruszak, Matylda Sauermann, Maciej Wołczek, Bartosz Aniśko.

**Writing – review & editing:** Paulina Kowalewska, Małgorzata Wójcik, Aleksandra Banaszak, Kacper Bernatowicz, Mateusz Motyl, Patrycja Wołodźko, Eryk Pietruszak, Matylda Sauermann, Maciej Wołczek, Bartosz Aniśko.

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
