## [Decision Letter · Decision Letter 0]

19 Nov 2025

Dear Dr. Kowalewska,

Thank you for submitting your manuscript to PLOS ONE. After careful consideration, we feel that it has merit but does not fully meet PLOS ONE’s publication criteria as it currently stands. Therefore, we invite you to submit a revised version of the manuscript that addresses the points raised during the review process.

**ACADEMIC EDITOR:** mechanisms underlying the links between cervical spine pain and postural disorders, and therefore the impact on physical activity levels, are insufficiently explained==============================

We look forward to receiving your revised manuscript.

Kind regards,

Thomas Rulleau, PT PhD

Academic Editor

PLOS ONE

Journal Requirements:

4. Please ensure that you refer to Figures 4, 5, 6 and 7 in your text as, if accepted, production will need this reference to link the reader to the figure.

5. Please note that your Data Availability Statement is currently missing a direct link to access each database]. If your manuscript is accepted for publication, you will be asked to provide these details on a very short timeline. We therefore suggest that you provide this information now, though we will not hold up the peer review process if you are unable.

Additional Editor Comments:

Please respond to the reviewers' comments to improve the readability and impact of your work.

Reviewers' comments:

Reviewer's Responses to Questions

**Comments to the Author**

1. Is the manuscript technically sound, and do the data support the conclusions?

Reviewer #1: Yes

Reviewer #2: No

2. Has the statistical analysis been performed appropriately and rigorously?

Reviewer #1: Yes

Reviewer #2: No

3. Have the authors made all data underlying the findings in their manuscript fully available?

Reviewer #1: Yes

Reviewer #2: Yes

4. Is the manuscript presented in an intelligible fashion and written in standard English?

Reviewer #1: Yes

Reviewer #2: No

Reviewer #1: This paper examines the relationships between spinal pain, physical activity, and functional performance in older adults.

Twenty-eight community-dwelling seniors (25 women and 3 men, mean age 70.8

years) participated in the study.

Despite the small sample size, the study provides interesting arguments in favor of preventive measures against pathological aging and physical inactivity.

I have no major objections to the acceptance of the manuscript for publication.

But the mechanisms underlying the links between cervical spine pain and postural disorders, and therefore the impact on physical activity levels, are insufficiently explained.

Ex : -Correlation between cephalic repositioning error and postural instability in elderly people.

Reddy, R. S., Alkhamis, B. A., Kirmani, J. A., Uddin, S., Ahamed, W. M., Ahmad, F., Ahmad, I., & Raizah, A. (2023). Age-Related Decline in Cervical Proprioception and Its Correlation with Functional Mobility and Limits of Stability Assessed Using Computerized Posturography: A Cross-Sectional Study Comparing Older (65+ Years) and Younger Adults. Healthcare (Basel, Switzerland), 11(13), 1924.

-Links with risk of falling and frailty

Uthaikhup S, Jull G, Sungkarat S, Treleaven J. The influence of neck pain on sensorimotor function in the elderly. Arch Gerontol Geriatr. 2012;55(3):66772.

Kendall JC, Hvid LG, Hartvigsen J, Fazalbhoy A, Azari MF, Skjødt M, et al. Impact of musculoskeletal pain on balance and concerns of falling in mobility-limited, community-dwelling Danes over 75 years of age: a cross-sectional study. Aging Clin Exp Res. 2018;30(8):969–75.

Reviewer #2: Introduction

- Literature update required: The reference cited on line 61 is more than ten years old, which is not appropriate for a public health topic that has been extensively investigated in recent years.

- Lack of transitions: Several sections would benefit from clearer and more explicit transitions to improve overall coherence.

- Scientific justification: The research gap is not clearly articulated and requires a more explicit formulation to justify the need for the study.

Methods

- The study design is not reported.

- The participant recruitment procedure is not described.

- The table presenting patient characteristics belongs to the Results section. Moreover, it should include data related to physical tests, pain indices, etc., in order to provide a complete description of the sample across all outcomes.

- The exclusion criteria listed are actually non-inclusion criteria; it is important to make this semantic distinction and revise the terminology accordingly.

- It would be more logical to present the inclusion and non-inclusion criteria before the protocol elements described on lines 120–121.

- Psychometric properties of the tests and questionnaires are missing (no information about their validity or reliability is provided).

- The postural stability outcome derived from stabilometry is not the most relevant. Although used in research, mean sway velocity of the center of pressure is more appropriate in geriatric populations; a static analysis with this parameter would be preferable.

- The TUG does not measure gait speed, which is typically assessed over 10 meters. This outcome is essential to characterize the functional level of the sample and should appear in the descriptive table.

- The visuomotor reaction test is interesting but uncommon; additional information is needed regarding its validity for older adults.

- No justification is provided for the sample size (n = 28); no power analysis or sample size calculation is mentioned.

- Statistical analyses: Several collected variables are not exploited. No multivariate analysis or adjustment for covariates (e.g., age, muscle mass) is presented.

Results

- A full descriptive section of the study population is required.

- Several interpretations are included in the Results section; these should be moved to the Discussion.

- The correlation between age and step length is not relevant to the stated research question, which focuses on pain and physical function in older adults; the same applies to other reported correlations.

- For Figure 2, the statistical values (r and p) reported in the text differ from those shown in the figure legend.

- No summary table of correlations is provided. Such a table would be more synthetic and readable than seven separate figures.

Discussion

- The discussion describes correlational findings as though they were causal (e.g., line 305).

- Several long paragraphs cite references without clearly linking them to the study’s findings (e.g., sections on gait or compensatory strategies).

- The hypotheses are never explicitly stated in the article, yet the discussion claims they are confirmed.

**Do you want your identity to be public for this peer review?** For information about this choice, including consent withdrawal, please see our Privacy Policy

Reviewer #1: No

Reviewer #2: No

<qb-div data-qb-element="re-enable-flow" style="z-index: 2147483647; max-width: 1px; max-height: 1px; box-sizing: border-box; position: fixed; top: 10px; right: 10px;"></qb-div>

---

## [Author Response · Author response to Decision Letter 1]

19 Dec 2025

1. Academic Editor

AE-1

Comment: “mechanisms underlying the links between cervical spine pain and postural disorders, and therefore the impact on physical activity levels, are insufficiently explained.”

Response:

We agree and have expanded the Discussion to provide a clearer, evidence-based mechanistic rationale linking cervical pain to postural control disturbances and, downstream, to potential reductions in physical activity.

• Added a distinct mechanistic subsection describing how cervical pain and sensorimotor deficits can impair cervical proprioception and multisensory integration (cervical-visual-vestibular), contributing to increased postural sway and fall risk.

• Explicitly connected this framework to our data, including the observed association between neck disability and COP sway area.

• Clarified the plausible behavioral pathway to physical activity (pain-related avoidance, reduced confidence, and fear of falling), while emphasizing that our cross-sectional findings support associations rather than causation.

• Strengthened the interpretation of the IPAQ results by acknowledging self-report limitations and explaining how these limitations may influence observed relationships.

Overall, these additions ensure that cited studies are directly linked to our specific outcomes and that the mechanistic argument is presented as an explanatory model, not a causal conclusion.

Manuscript changes:

Discussion: new/expanded mechanistic subsection (cervical pain - sensorimotor deficits - postural sway) and an explicit paragraph linking the framework to physical activity behavior and IPAQ interpretation; Limitations: strengthened language on cross-sectional design and self-report physical activity.

2. Reviewer 1

R1-1

Comment: The reviewer considers the work interesting and valid but asks for clearer mechanisms linking neck pain to postural disorders and physical activity, with updated references.

Response:

We thank the reviewer. Our revisions are aligned with the response to the Academic Editor and focus on strengthening the mechanistic explanation and ensuring that the literature is current and directly relevant to our findings.

• Expanded the Discussion with a structured description of the cervical sensorimotor deficit model in older adults.

• Updated and integrated key mechanistic references and linked them explicitly to our outcomes (notably the NDI - COP sway area association).

• Clarified the plausible connection to physical activity behavior and framed it cautiously (association, not causation).

Manuscript changes:

Discussion: mechanistic subsection expanded and updated; Introduction/Discussion: references refreshed where prior citations were outdated.

3. Reviewer 2

3.1 Introduction

R2-1

Comment: Some cited literature (particularly public health) is older than 10 years and should be updated.

Response:

We updated the Introduction to include more recent evidence (including population-based studies and meta-analytic work) on:

• population aging trends;

• the epidemiology of spinal pain (cervical and lumbar);

• the relationship between physical activity, functional fitness, and health in older adults.

Older references were retained only when they represent foundational work, and they are now complemented by current sources.

Manuscript changes:

Introduction: updated epidemiology/public-health background paragraphs and updated reference list.

R2-2

Comment: Lack of transitions.

Response:

We improved the flow of the Introduction by adding clear transitional sentences:

• between the physical activity background and the introduction of pain as a barrier to movement;

• between risk-factor discussion and aging/fitness context;

• between the background and the study aim;

• between the aim and hypotheses.

Manuscript changes:

Introduction: revised transitions and reorganized paragraph order for clearer logic.

R2-3

Comment: Scientific justification / research gap not clearly articulated.

Response:

We rewrote the final part of the Introduction to make the research gap explicit and to clarify why this study adds value. In particular, we now emphasize the limited number of studies that simultaneously evaluate spinal pain (cervical and lumbar) alongside balance, mobility, reaction time, step length, muscle strength, and physical activity in community-dwelling older adults.

Manuscript changes:

Introduction: revised final paragraph (research gap) and strengthened study rationale.

R2-4

Comment: Hypotheses not explicitly stated in the article.

Response:

We now state the hypotheses explicitly at the end of the Introduction. We also adjusted wording throughout the manuscript to refer to results as being 'consistent with' hypotheses rather than 'confirming' them.

Manuscript changes:

Introduction: hypotheses added explicitly; Discussion: wording adjusted to avoid confirmatory/causal language.

3.2 Methods

R2-5

Comment: Study design is not reported.

Response:

We now explicitly state at the beginning of the Methods that this is an observational, cross-sectional study, and we note that reporting follows STROBE guidance.

Manuscript changes:

Methods: opening paragraph updated to specify study design and reporting standard.

R2-6

Comment: Participant recruitment procedure is not described.

Response:

We expanded the Participants subsection to detail recruitment channels (e.g., senior clubs, local community announcements) and the screening approach used to verify eligibility.

Manuscript changes:

Methods - Participants: recruitment and screening details added.

R2-7

Comment: Table with characteristics belongs to Results and should include all outcomes.

Response:

We moved the participant characteristics table to the Results section and expanded it into a comprehensive Table 1 including demographics and all clinical/functional outcomes.

Manuscript changes:

Results: Table 1 placed in Results and expanded (demographics + all outcomes).

R2-8

Comment: Exclusion criteria vs non-inclusion terminology.

Response:

We standardized terminology throughout the manuscript by using only 'inclusion criteria' and 'exclusion criteria' and removed the ambiguous term 'non-inclusion'.

Manuscript changes:

Methods: eligibility terminology standardized and clarified.

R2-9

Comment: Order of inclusion/exclusion criteria vs protocol description.

Response:

We reorganized the Methods so that eligibility criteria are presented before the detailed testing protocol and procedures.

Manuscript changes:

Methods: reordered subsections for clearer readability.

R2-10

Comment: Psychometric properties of tests and questionnaires are missing.

Response:

We added a dedicated paragraph in the Methods summarizing validity/reliability and key measurement considerations for each tool. In addition, we acknowledged known limitations where relevant (e.g., self-report bias and overestimation in IPAQ).

Manuscript changes:

Methods: new paragraph summarizing psychometric properties/measurement considerations for NDI/ODI, IPAQ, TUG, posturography outcomes, dynamometry, and reaction time testing.

R2-11

Comment: Postural stability outcome: mean sway velocity vs sway area.

Response:

We agree that mean sway velocity is an informative parameter. In this study we used sway area (ellipse area) as the primary stability outcome based on its interpretability and the available reliability evidence in older adults. We now justify this choice explicitly and list the absence of velocity measures as a limitation and recommendation for future work.

Manuscript changes:

Methods: rationale for sway area added; Discussion/Limitations: explicit note on omission of sway velocity and recommendation for future studies.

R2-12

Comment: TUG does not measure gait speed; 10 m walk test is essential.

Response:

We agree that TUG reflects functional mobility and dynamic balance rather than isolated gait speed. We clarified this in the Methods and removed statements implying that TUG measures gait speed. We also acknowledge the absence of a dedicated gait-speed test (e.g., 10 m walk test) as a limitation.

Manuscript changes:

Methods: clarified construct measured by TUG; Limitations: absence of 10 m walk test/gait-speed metric noted.

R2-13

Comment: Visuomotor reaction test: provide more information on validity in older adults.

Response:

We added a clearer description of the reaction-time assessment (task, outcome definition, and device/system used). Where validation evidence in older adults is available, we cite it; where evidence is limited, we acknowledge this explicitly as a measurement limitation.

Manuscript changes:

Methods: expanded reaction-time testing description; Limitations: clarified evidence base and potential measurement constraints.

R2-14

Comment: No sample size calculation / power analysis.

Response:

We acknowledge that no a priori sample-size calculation was conducted. To address the concern without overstating the role of post-hoc calculations, we added a brief sensitivity statement indicating the approximate correlation magnitude that can be detected with N = 28 at alpha = 0.05 (two-tailed). We also strengthened the limitations regarding generalizability.

Manuscript changes:

Statistical analysis: added sensitivity/power statement; Limitations: expanded sample-size/generalizability limitation.

R2-15

Comment: No multivariate analysis / no adjustment for covariates.

Response:

We agree that covariate-adjusted models would be valuable. However, with N = 28, multivariable regression with several predictors/covariates would be statistically unstable and at high risk of overfitting. We therefore limited analyses to bivariate correlations and clearly stated this rationale, while acknowledging it as a limitation.

Manuscript changes:

Statistical analysis: clarified rationale for bivariate approach; Limitations: lack of covariate adjustment explicitly noted.

3.3 Results

R2-16

Comment: A full descriptive section of the study population is required.

Response:

We added a descriptive paragraph summarizing participant characteristics and direct references to the comprehensive Table 1.

Manuscript changes:

Results: descriptive paragraph added; Table 1 referenced appropriately.

R2-17

Comment: Results contain interpretations; move to Discussion.

Response:

We removed interpretive language (e.g., 'this suggests', 'indicating that') from the Results. The Results now report data only; interpretations were moved to the Discussion.

Manuscript changes:

Results: interpretive phrasing removed; Discussion: interpretation consolidated.

R2-18

Comment: Relevance of correlations (age-step length, etc.) to the main research question.

Response:

We clarified that correlations involving age (e.g., age vs step length/reaction time) are secondary analyses intended to characterize the functional profile of the sample rather than the primary focus of the pain analysis.

Manuscript changes:

Introduction/Discussion: clarified primary vs secondary aims and the role of age-related correlations.

R2-19

Comment: Inconsistencies between r and p values in text vs figures.

Response:

We verified all calculations and ensured that all r and p values are consistent across the text and the revised correlation table. Figures that previously contained inconsistencies were removed.

Manuscript changes:

Results: values harmonized with Table 2; figures removed and cross-references cleaned.

R2-20

Comment: No summary table of correlations; seven figures instead.

Response:

We removed the multiple correlation figures and replaced them with a single Spearman correlation matrix (Table 2), which presents the results more transparently.

Manuscript changes:

Results: Figures removed; Table 2 added and cited in the text.

3.4 Discussion

R2-21

Comment: Correlations described as causal.

Response:

We agree this is important. We carefully revised the Discussion to remove causal wording (e.g., 'influences', 'affects') and replaced it with associative language (e.g., 'associated with', 'co-occurs with', 'consistent with'). We also added an explicit statement that the findings represent associations, not causal effects.

Manuscript changes:

Discussion: causal language removed; explicit association-only statement added.

R2-22

Comment: Long paragraphs with references not clearly linked to the findings.

Response:

We streamlined the Discussion by splitting long paragraphs and ensuring that each cited study is directly tied to our specific results or to the mechanistic explanation of those results.

Manuscript changes:

Discussion: paragraph structure improved; references better aligned with study findings.

3.5 Editorial and formal requirements

We addressed all remaining editorial and formal requirements as follows:

• Adjusted manuscript structure and file naming to match journal templates.

• Ensured the ethics statement is placed exclusively in the Methods section.

• Updated the corresponding author's ORCID in the submission system.

• Removed references to deleted figures and ensured new tables are correctly cited.

• Provided a direct DOI to the public repository for data availability.

• Verified the reference list (including adding suggested literature and checking for retractions, where applicable).

• Revised the manuscript for clarity and standard English usage (grammar and typography).

We again thank the Academic Editor and Reviewers for their time and guidance. We hope the revised manuscript adequately addresses all concerns and is now suitable for publication.

Sincerely,

Paulina Kowalewska, on behalf of all authors

---

## [Decision Letter · Decision Letter 1]

21 Jan 2026

Dear Dr. Kowalewska,

Thank you for submitting your manuscript to PLOS ONE. After careful consideration, we feel that it has merit but does not fully meet PLOS ONE’s publication criteria as it currently stands. Therefore, we invite you to submit a revised version of the manuscript that addresses the points raised during the review process.

minor revisions are required to ensure full compliance with the journal’s presentation guidelines:

1. the addition of a brief introductory sentence in the abstract, prior to the statement of the objective;

2. the inclusion of captions below the figures in the Results section, along with explicit references to these figures within the main text.

We look forward to receiving your revised manuscript.

Kind regards,

Thomas Rulleau, PT PhD

Academic Editor

PLOS One

Journal Requirements:

Additional Editor Comments :

minor revisions are required to ensure full compliance with the journal’s presentation guidelines:

the addition of a brief introductory sentence in the abstract, prior to the statement of the objective;

the inclusion of captions below the figures in the Results section, along with explicit references to these figures within the main text.

Reviewers' comments:

Reviewer's Responses to Questions

**Comments to the Author**

Reviewer #1: All comments have been addressed

Reviewer #2: All comments have been addressed

2. Is the manuscript technically sound, and do the data support the conclusions?

Reviewer #1: Yes

Reviewer #2: Yes

3. Has the statistical analysis been performed appropriately and rigorously?

Reviewer #1: Yes

Reviewer #2: Yes

4. Have the authors made all data underlying the findings in their manuscript fully available?

Reviewer #1: Yes

Reviewer #2: Yes

5. Is the manuscript presented in an intelligible fashion and written in standard English?

Reviewer #1: Yes

Reviewer #2: Yes

Reviewer #1: (No Response)

Reviewer #2: The reviewer notes that all previous recommendations have been fully addressed and commends the authors for the substantial work undertaken. This revised version (V2) represents a clear added value compared to the initial submission (V1), both in terms of content and overall presentation. The manuscript has been significantly improved and now demonstrates greater clarity, coherence, and scientific quality.

The methodological section is particularly well constructed and clearly presented, strengthening the robustness and reproducibility of the study. In addition, the inclusion of more recent and relevant references in the introduction is appreciated, as it better situates the study within the current state of the literature.

The discussion is also notably improved and offers new and thoughtful insights, enhancing the interpretation and impact of the results.

Only very minor revisions are required to ensure full compliance with the journal’s presentation guidelines:

the addition of a brief introductory sentence in the abstract, prior to the statement of the objective;

the inclusion of captions below the figures in the Results section, along with explicit references to these figures within the main text.

Subject to these minor adjustments, the manuscript is considered satisfactory.

**Do you want your identity to be public for this peer review?** For information about this choice, including consent withdrawal, please see our Privacy Policy

Reviewer #1: No

Reviewer #2: No

---

## [Author Response · Author response to Decision Letter 2]

25 Jan 2026

We would like to thank the Academic Editor and the Reviewers for their careful evaluation of our manuscript and for the constructive comments provided.

We have addressed all remaining minor comments as follows:

1. Abstract

A brief introductory sentence has been added at the beginning of the abstract, prior to the statement of the study objective, as requested.

2. Figures and tables in the Results section

In accordance with the previous recommendations and to improve clarity and readability, the figures in the Results section have been replaced with tables. The relevant results are now presented in tabular form and are explicitly referenced in the main text.

We believe that these revisions ensure full compliance with the journal’s presentation guidelines.

We appreciate the positive assessment of our work and hope that the revised manuscript is now suitable for publication in PLOS ONE.

Kind regards,

Paulina Kowalewska

---

## [Decision Letter · Decision Letter 2]

19 Feb 2026

Physical function, daily activities, and spinal pain in the elderly: A cross-sectional study

PONE-D-25-51057R2

Dear Dr. Kowalewska,

We’re pleased to inform you that your manuscript has been judged scientifically suitable for publication and will be formally accepted for publication once it meets all outstanding technical requirements.

Kind regards,

Yih-Kuen Jan, PhD

Academic Editor

PLOS One

Additional Editor Comments (optional):

Reviewers' comments:

Reviewer's Responses to Questions

**Comments to the Author**

Reviewer #2: All comments have been addressed

2. Is the manuscript technically sound, and do the data support the conclusions?

Reviewer #2: Yes

3. Has the statistical analysis been performed appropriately and rigorously?

Reviewer #2: Yes

4. Have the authors made all data underlying the findings in their manuscript fully available?

Reviewer #2: Yes

5. Is the manuscript presented in an intelligible fashion and written in standard English?

Reviewer #2: Yes

Reviewer #2: Les reviewers apprécient les nouvelles modifications apportées qui permettent un article bien construit, intelligible et en respect avec les attendus de la revue

**Do you want your identity to be public for this peer review?** For information about this choice, including consent withdrawal, please see our Privacy Policy

Reviewer #2: No

---

## [Editor Report · Acceptance letter]

PONE-D-25-51057R2

PLOS One

Dear Dr. Kowalewska,

I'm pleased to inform you that your manuscript has been deemed suitable for publication in PLOS One. Congratulations! Your manuscript is now being handed over to our production team.

Kind regards,

on behalf of

Dr. Yih-Kuen Jan

Academic Editor

PLOS One